# Research Ethics Challenges, Controversies and Difficulties in Intensive Care Units—A Systematic Review of Theoretical Concepts

**DOI:** 10.3390/nursrep15050164

**Published:** 2025-05-07

**Authors:** Cristina Petrișor, Mara Chirteș, Tudor Magdaș, Robert Szabo, Cătălin Constantinescu, Horațiu Traian Crișan

**Affiliations:** 1Anesthesia and Intensive Care II Department, “Iuliu Hațieganu” University of Medicine and Pharmacy, 400012 Cluj-Napoca, Romania; mara.chirtes@elearn.umfcluj.ro (M.C.); tudor.miha.magdas@elearn.umfcluj.ro (T.M.); szabo.robert@elearn.umfcluj.ro (R.S.); constantinescu.catalin@umfcluj.ro (C.C.); 2Medical Education Department, “Iuliu Hațieganu” University of Medicine and Pharmacy, 400012 Cluj-Napoca, Romania; horatiu.crisan@umfcluj.ro

**Keywords:** research ethics, intensive care, ethical challenge, nursing, research nurse

## Abstract

**Background:** Research in the intensive care unit (ICU), which involves critically ill patients, families and healthcare professionals, poses significant ethical challenges. The aim of this PRISMA-guided systematic review is to identify major challenges for research ethics in the ICU. **Methods:** Pubmed and Scopus databases were searched in November-December 2024 for papers discussing theoretical concepts or specific aspects related to ethical issues in ICU research, retaining 70 papers on ICU research challenges, difficulties or controversies. **Results:** A total of 10 papers described general concepts related to research ethics in the ICU, 16 studies focused on the methodology or some of the study steps, and 6 papers focused on ICU trials, while 38 studies focused on special patient categories or special situations of critical patients. None of the included papers addressed all of the issues we identified regarding the ethical challenges. **Conclusions:** ICU research is challenging from a moral point of view. Significant ethical difficulties arise during the design and implementation phases, hampering the study's exactness. Being a vulnerable population with limited decision-making capacity and research-associated risks, alternative consent methods need to be considered. This systematic review provides a checklist of aspects that could generate ethical dilemmas and might constitute a starting point in the conduct of research studies, preventing unethical research.

## 1. Introduction

Research in the intensive or critical care unit (ICU), addressing critically ill patients, families or healthcare professionals, poses significant ethical challenges due to the high complexity of the life-threatening conditions of the patients and the environment in which the research is carried out.

ICUs are situated at the crossroads between different medical specialities and different patient categories. ICUs are organised around an inhomogeneous population of critically ill patients with organ dysfunctions or insufficiencies requiring continuous monitoring and invasive interventions to survive while curative treatments are applied. Here, novel technologies and innovative therapies are constantly being searched for and developed [1]. The requirement for prompt responses to sudden changes in patient conditions leaves little room for research activities. The optimal treatments and care for the critically ill are yet to be achieved, creating pressure for those caring for patients, especially when definitive answers or consensus are lacking. ICUs admit a high number of patients, from whom high volumes of complex data are gathered. Hospitals handle data differently, making it difficult to standardise research methods across institutions [2,3]. The demanding nature of ICU settings presents significant challenges for research.

Critically ill patients require support for vital functions as they have complex physiological derangements, clinical presentations and organ dysfunctions [4]. This wide heterogeneity produces a lack of consensus, which makes it difficult to standardise protocols or generalise findings [5,6]. In turn, this affects research. Decisional capacity is also affected, making these patients even more vulnerable. The acute disease, altered mental status, encephalopathy, and medications preclude informed consent, passing this responsibility to the medical team together with the surrogate [7]. The family might face intense emotional challenges, so patient recruitment for clinical studies has some limitations.

Should research be conducted at all in the ICU, where the sickest of the patients are admitted? It is generally accepted that research should be conducted in vulnerable populations, and despite the many challenges, research is essential to improve survival, quality of life, and lower complication rates [8].

There is a general tendency to introduce procedures to clinical practice without rigorous evaluation in critical care [9]. Is one of the reasons for this the difficulty in conducting rigorous studies? Truog et al. stated that “if current trends continue, within several years it could become nearly impossible to conduct research in critical care medicine”, pointing out the difficult informed consent process in this vulnerable population [10]. Bigatello et al. stated in 2003 that there was a lack of guidelines for the ethical conduct of research in the ICU [4]. This could lead to unethical research practices in the ICU. Designing and implementing guidelines is only possible if ethical challenges or difficulties are identified and described.

The aim of this systematic review is to identify major challenges, difficulties, controversies and particularities for research ethics in the ICU, and to provide a list of key questions, items or situations relevant for the design of research involving critically ill patients. These might help researchers anticipate and identify potential barriers and limitations in the conduct of research studies.

## 2. Materials and Methods

Pubmed and Scopus databases were searched for “Research ethics AND ICU”. Our search revealed 2817 papers in Pubmed and 12,276 papers in Scopus. Papers that discussed theoretical concepts or specific aspects related to ethical issues in ICU research domains or populations were screened by the investigators, selecting 315 papers from Pubmed and 5429 papers from Scopus based on title and abstract information.

From these, we excluded empirical studies, research based on observation and experience, and papers on clinical ethics, if they did not discuss aspects related to research ethics in the ICU. We included in this PRISMA-guided systematic review theoretical papers with argumentative patterns and other papers discussing challenges, difficulties or controversies related to the ethical conduct of research studies in the ICU. No formal quality assessment was performed as the included papers were theoretical ones. A total of 70 theoretical papers based on the initial search and an analysis of the references of the included papers were retained in the final analysis.

From the 70 included papers, 10 described general concepts related to research ethics in the ICU, 16 studies focused on the methodology of the research studies or some of the study steps, 6 papers focused on ICU trials, while 38 studies focused on special patients’ categories or special situations of critical patients (Figure 1. Prisma flow diagram for review methodology) [11].

## 3. Results

The first manuscript to address ethical challenges in ICU settings is a paper by Lamdin et al. on research involving severely and critically ill patients [12]. By the year 2000, this topic had rarely been addressed. Afterwards, the number of publications increased in time, with genomic research concerns and pandemic-related ethical issues being increasingly recognised since 2020 (Figure 2).

The published papers have been classified differently as manuscript types: 42 original articles (60%), 11 reviews (15.71%), 6 editorials (8.57%), while others are commentaries, proceeding papers, letters and viewpoints. Six papers (8.57%) have not been classified as a specific type of manuscript (Figure 3).

The ten general papers on the conduct of ethical research in the ICU approached the main ethical principles of research [1,4,8,9,10,12,13,14,15,16], but also included discussions on the steps of research studies that might generate moral debates when studying the critically ill population, certain special populations that could represent the research study subjects in the ICU and some research settings related to ICU practice. None of the ten general papers addressed all the issues we identified regarding the ethical challenges.

From the 16 papers addressing the study steps [7,17,18,19,20,21,22,23,24,25,26,27,28,29,30,31], which reflect the design and methodology of the planned research studies, most had patient consent as a main theme, while others focused on participation in research, risk of research, recruitment and data sharing. At the same time, some of these theoretical papers focused on more than one or two study steps and discussed the ethical challenges of research more comprehensively. A total of six papers specifically addressed the ethics of clinical trials conducted in the ICU, describing their methodology and limitations [32,33,34,35,36,37].

A total of 38 papers focused on specific ethical challenges of ICU research involving special patient populations or special settings of ICU care, organisation or costs (Table 1).

## 4. Discussion

Currently, large controlled studies including critically ill patients are lacking, and many patients are treated based on knowledge transferred from non-critically ill patients [15]. The question of what to investigate in the ICU remains, as there are no definitive answers on the outcome of certain diseases. Triage decisions are not uniform worldwide. Complication rates and the success of organ support measures still need to be investigated. Moreover, psychological burden, burnout and staff depression, organisational aspects and healthcare-associated harms must be discussed. In such situations, a clinical question regarding the optimal care should be followed by adequately designed studies that respect ethics involving humans and that are mandatory to win public trust [9,14]. Challenges in research ethics must be identified to conduct ethical clinical research by taking into account considerations such as social value, scientific validity, respect for autonomy, privacy, dignity and minimal harm.

Clinical research is essential for safe care, improving outcomes, department organisation and staff wellbeing. Lack of ICU research leads to a lack of knowledge that impacts clinical care. Patients, families and staff need to be involved to understand their roles, as cooperation is necessary [15]. The ICU environment creates many research opportunities. What are the best ways to achieve knowledge regarding the critically ill population? How should researchers be guided regarding the ethical conduct of research?

In 1974, Lamdin et al. presented the conclusions of the American Heart Association, aiming to bring to attention ethical issues that deserve consideration. The authors stated, “No previous discussion has been addressed to seriously and critically ill patients” [12]. Considering this as the first description of the attention for the ethical conduct of research in ICUs, we can state that for a long time, the medical literature has been addressing topics of research ethics in the ICU. We identified approximately 70 theoretical papers on the topic.

Furthermore, a second paper discussing the ethical aspects related to ICU clinical research presented the ideas expressed in a medical conference and provided recommendations, emphasising the necessity and priority of such studies [16].

These two reference papers address the ethical concepts globally and are the two most comprehensive, but neither of the papers included in this systematic review addresses all potential challenges. For sure, given the particularities of the critically ill population and the ICU environment, critical care research imposes special ethical challenges and dilemmas, which are sometimes difficult to anticipate [1,4,8].

To conduct ethical research in the ICU safely, investigators need to have in-depth knowledge of the difficulties of clinical research involving critically ill patients. We conducted this systematic review, after a comprehensive search of the medical literature, in order to identify potential barriers, challenges and controversies in ICU research. In this review, we have gathered key topics or issues regarding the ethical conduct of research in the ICU, aiming to provide intensive care researchers a checklist of potential elements or particularities that might lead to ethical dilemmas or predispose to unethical conduct of research in the ICU. Thus, researchers could anticipate potential barriers that need to be considered during the design, implementation, monitoring and publication of ICU-related research studies. By conducting ethical research, conclusions could be applied to clinical practice, both domains having similar ethical principles. The ethical controversies or challenges were categorised into those addressing the study steps and methodology, special populations and special settings in ICU care.

I.
**
*What are the moral principles that govern ethical research in the ICU?*
**


Ethical research is the prerequisite for valid answers to questions regarding the critically ill population. This would allow optimal care for the ICU patient, improve morbidity and lower mortality rates, and also gain public or social trust in the medical profession, ICU physicians and researchers. This could be achieved by addressing treatments specific for the ICU, like mechanical ventilation or other organ replacement techniques. The ethical integrity of the research must be ensured at all steps of the process: research question posed, study design, consent process, research oversight, data management and analysis, interpretation and dissemination of results [16].

Researchers must consider that the ICU environment also has the ethical particularities of clinical practice; communication problems, limited understanding, informed consent conditioning, beneficence, autonomy, increased technology, variability in practice, and advanced planning questions are everyday concerns [74]. Clinical research is distinguishable from clinical care [8]. Still, the practice of intensive care medicine as a medical speciality and ICU research are interconnected, and one could not exist without the other. Clinical practice is ethical only when it has been proven through research [14]. Conducting research in the ICU setting is a moral duty. Otherwise, failure to improve outcomes through rigorous studies is as ethically irresponsible as failing to provide care [13]. The boundaries between clinical research and clinical practice are sometimes unclear [4]. However, research practice is different from clinical practice, and the investigators should ideally not be involved in the clinical management of the patients.

To achieve ethical conduct of research, all the required steps in the design and conduct of research are important and should respect the biomedical principles. Autonomy (which is related to respect for each individual person), beneficence, non-maleficence and justice are fundamental moral values of the medical profession [75]. These clinical ethical desiderata are also mentioned as key moral duties for ICU research, demonstrating their universality. Ethical principles were not originally intended to be applied to research and were not conceived to meet the challenges of ICU research [1]. The advantage of having similar moral principles in clinical practice and research is that the best interest of the patient is the ultimate desideratum according to the “Do no harm!” principle. In fact, the moral principle of beneficence, together with its opposite, non-maleficence, might generate a conflict. These concepts apply to both society and larger groups of critically ill patients, as well as to the individual patient who becomes the study subject and might suffer from harm as a result of study risk. In research, potential social value should be compared to individual risks.

Ethical principles guiding research have long been established by ethicists, but are less often discussed among critical care investigators and less well highlighted in specialty journals [16]. The description of these moral values guiding research has been mentioned in six out of the seventy included papers [1,4,8,9,14,16]. Their highlighting might increase awareness and guide investigators in ICU research practice.

II.
**
*Any research study starts from the research question, which leads to the study hypothesis. Sometimes, the primary endpoint or goal has additional secondary goals, which represent secondary information that might be gained from the study*
**


The research question should be relevant to the critically ill, investigating aspects that are specific to this population and for which the scientific answer cannot be obtained by studying other patient categories, who might not be so severely ill. Examples are diagnostic methods, monitoring or treatments like hemodynamic invasive monitoring, mechanical ventilation, and complications of ICU-related care. The study question represents a hypothesis that the study tests, and the intention is to generate an acceptable scientific conclusion that might improve the outcome. The results of the clinical study should be generalisable to the ICU population [8]. This can only be achieved by designing a sound research project, starting from a research question for which the answer is not yet established within the scientific community (i.e., there is genuine clinical equipoise) [8,9,10,16,27].

The study is not designed for the benefit of the patient, but for the large category of patients to which the research subject belongs. Thus, it is not an individual good, underlining the social value of research.

Even if the study is designed so that generalisable results are intended for the benefit of society, the non-maleficence duty is strengthened, and the patient needs to be protected during research [1,4,7,8,9,10,12,13,14,15,16,17,24]. The balance between benefits and harms, i.e., a favourable risk-benefit profile of the study, needs to be considered as a measure of protection. Safeguarding the subjects during clinical studies is an ethical obligation for all participants in research: investigators, research ethics committees and sponsors [9]. It is easier to conduct no or low-risk studies, or to define adequately what acceptable risk means in the setting of critical care in order to safeguard against anticipated risks [4,9,10,12,13,16,20]. Since the studies are designed to have social value, liability in case of harm to the research subject is also a topic seldom addressed in theoretical papers on ICU research [1,12]. In order to avoid potential harm in a large number of patients, safety and efficacy could be first tested in a small number of patients in pilot studies [4]. During the recruitment phase, monitoring for adverse events is mandatory and is a duty of the researcher, sponsor and Research Ethical Board (REB). They have the ability to stop the study should there be any safety concerns. It is mandatory during the design phase to establish such safeguards.

The scientific validity of the study depends on optimal study methodology, enrollment criteria that fit with the scientific goals and outcome measures that reflect the purpose of the study, generating a sound research project [15,16].

When all these ethical desiderata, which in fact are moral requests for the study, are fulfilled, then results arising from ICU research have the premises to be valid.

III.
**
*The clinical researcher needs knowledge on critical care, communication abilities, critical thinking when elaborating the study hypothesis, monitoring the study and analysing results, as well as knowledge of ethics and law*
**


If the clinical researcher is also the treating physician of the patient, there is a concern over a conflict of interest. The researchers’ aim is to conduct the study, while the clinicians’ aim is to treat the patient. Critical care research and practice differ as the clinicians’ primary interest is to benefit individual patients, not the social value of the study [16].

IV.
**
*Conflicts of interest*
**


Conflicts of interest may occur at any stage of the research study and belong to any individual involved in the process of clinical research and clinical care [16].

Sometimes, the study design is rigid with allocation of the patients into one arm, while clinical practice is constantly adapted to the individual needs of the patient. Ideally, the treating physician is not the researcher who included the patient in the research study [8,12,16,22]. Other conflicts might arise between different members of the team, different medical and surgical specialities, or between the family and the medical team. For instance, in non-consensual research, the research team and the family might have different views regarding the enrollment of a patient in a research study [76]. Also, the research study might be perceived as an extra burden on the already difficult task of caring for a critical patient, and the recruitment process might be hampered [77]. Ideally, the study team should not place additional burdens or requests on the ICU team [1].

Other possible conflicts of interests might be the financial ones, with management and industry pressure involved in the high costs for ICU care and the high costs for research [1,9,16] or the researcher conflict of interest, since most researchers belong to academic staff of different organisations, implying publication duties and academic competitiveness [4,14,16].

V.
**
*The oversight and approval of a research study to be conducted in the ICU is performed by the REB, the members of which also have ethical duties for the conduct of the study*
**


The standard requirements for the approval of clinical studies are similar to other types of studies conducted in non-critically ill populations, but the independent review of the protocol needs to involve researchers with expertise in ICU clinical studies or expert consultants who understand the problems very well [1,7,8,9,12,14,16,27]. A challenge for researchers is that independent REBs ask for different requirements for protocol approval [1,39]. The approval processes of local and national ethics committees need to be harmonised through a common protocol, especially for multi-national genetic studies [39]. However, this fact is not limited to ICU research but is a characteristic of research involving human subjects in general.

VI.
**
*Study registration*
**


Interestingly, study registration before commencement is only mentioned by Luce et al., possibly because until then, large-scale trials in the ICU population were not so often performed [16].

VII.
**
*Recruitment of research subjects*
**


Only after the study was approved by the REB and the trial had been prospectively registered could a researcher start to recruit patients. There is no consensus about the need to prospectively register epidemiological or observational studies databases or registries.

The request to obtain approval before recruitment starts represents an obvious ethical conduct of research and is one of the regulations for all studies involving human subjects. This is possibly the reason why the request has not been clearly mentioned in any of the theoretical papers on critical care research. The recruitment phase implies the fair selection of participants: patients should be included only if the study question cannot have an answer without enrolling critically ill patients [7,8,9,16]. Since these are the sickest of the patients possible, their participation should be requested only for studies designed specifically for the population they belong to. Patient fragility is a decrease in the capacity to tolerate adverse events in research, and critically ill patients might be most fragile [14]. These patients, therefore, require safeguards during research.

A bias during the recruitment phase could be therapeutic misconception, when individuals or relatives believe that the individual patient could have a therapeutic benefit from research [16,22]. For instance, the parents of a very sick child with limited chances of survival might be inclined to confirm participation in research, hoping that the new therapy might cure the disease. The research team needs to inform the family in an honest way that no evidence is in favour of the new treatment, and this is why the study is in fact conducted. Correct disclosure of information during the informed consent process is very important so that no false hopes are induced.

VIII.
**
*None should recruit patients in a study if the patient or study subject does not agree*
**


Informed consent is a prerequisite for research [14]. This is one of the universal requests of research ethics worldwide. Consenting to participate in research is the foundation of trust between the researcher and the patients, as within a signed contract, and also between healthcare systems, researchers and society. The attitudes and views of the general population towards research participation are that researchers provide incomplete information regarding risks and benefits, and that approximately 30% of the patients are not informed about their participation in research [18]. This view underlines the need for the participants to understand and to provide valid informed consent, one of the fundamental steps in gaining public trust in research. In its absence, the recruitment process is negatively affected.

The challenge of consent in critically ill patients has been extensively analysed in the literature, since patients are cognitively impaired, captive, dependent on care, they might lack decisional capacity, and are frequently too ill to understand [1,4,8,9,12,13,14,16,17,22,24,26,30]. In ICUs, most often, patients are not able to make decisions for themselves, and research participation is one such decision. It may be difficult to diagnose the loss of mental capacity in ICU patients [4]. Still, consent is a key step in patient protection and a proof of respect for autonomy that cannot be avoided in ICU research. Whoever signs the informed consent for research participation should act in the patient’s best interest [9,21]. The barriers and challenges in the inclusion process led to the omission of three out of four eligible critically ill patients, and these are generated by the issues related to autonomy and validity of consent, at least in part [7].

The ethical analysis of the informed consent process in the ICU emerges as a consequence of the inability of the patient to understand aspects related to research and provide valid consent. Frequently, patients are mechanically ventilated, comatose or sedated, and present cognitive dysfunction like delirium. Thus, any attempt to truly inform the patients might seem in vain. The particularities of the patients might prevent patients from participating in ICU research studies. Ideally, a uniformly accepted procedure for consenting critically ill patients should be adopted by all investigators [30]. To have such a procedure for this patient category might only be a dream.

To overcome the challenges of informing the patient and gaining consent, alternative consent methods have been proposed. In the absence of these, any study in the ICU might be almost impossible to conduct, as patients cannot give a truly informed consent.

The most frequent practice in ICU research is to ask for permission from a proxy or surrogate, for example, the family in most legislative systems. However, the family members might suffer from emotional distress and anxiety when a family member is critically ill and might not be able to comprehend the information. Also, it seems that the proxy’s ability to predict patients' preferences is highly variable [17]. There might be false positives and false negatives when the proxy appreciates the critical patients’ wish to participate in clinical research, and it seems that these assumptions are no better than tossing a coin [78]. There certainly are situations in which family members disagree. Informed surrogate consent is far from being ideal, as most do not know with certainty the position of the patient in such a situation [4]. Even if there are some question marks regarding surrogate consent, the positive attitudes and preferences of relatives of ICU patients can allow research to be conducted [23]. Having surrogate informed consent is better than having no consent for the recruitment of critically ill patients in research studies and can contribute to gaining family trust in the research team.

Alternative methods that might allow patient recruitment are the use of waivers of informed consent (only when risk is minimal), deferred, delayed or retrospective consent (after the patient becomes able to consent) [4,13,16,22,24,27,44,49]. Some authors proposed the involvement of communities in the consent process, providing information that is tailored to the patients' or family’s ability to comprehend and raising awareness towards ICU research [24,27]. Some authors argue that in special circumstances, depending on the risk the patient is exposed to as a research subject, if there is clinical equipoise on the research question, the emergency situation would not allow time for consent, the study is approved by the REB and safety monitoring is embedded in the design of the study, there might be exemptions to the rule of gaining written informed consent [27].

In the informed consent documents, researchers need to document what information about the study was provided and that study subjects or the surrogate can withdraw the patient from the study at any time. A written agreement to participate in research is mandatory.

IX.
**
*Since the recruitment phase, data protection is a duty*
**


Personal data and biological samples protection needs to be a point of attention for everybody involved in ICU research [14,15,16,29]. This might become very important as digitalisation and electronic data acquisition of real-time variables have emerged. As this type of data becomes more available worldwide, the researchers can work remotely. It is possible that this technological advance will lead to other ethical challenges in the future.

X.
**
*What type of study should the researcher choose to find the correct answer to ICU research questions?*
**


Choice of the type of studies that may be carried out may be difficult [12]. The standard to provide answers to research questions is the performance of randomised controlled trials (RCTs). In the ICU, these are difficult to perform because of clinical equipoise debates, difficulty in recruitment, and the high-risk patient profile who might not tolerate additional study-related risks. Truog et al. argue that there is unreasonable insistence on randomised controlled trials [10]. Observational studies, either prospective or retrospective, in which there is no attempt to modify treatments, have scientific value by providing evidence that allows the understanding of the prevalence, incidence, risk factors and prognosis of critical illness. These types of studies have the advantage that multicenter protocols can be designed and applied in several countries at the same time, providing information on variability in care [9,12,13,16].

XI.
**
*Conducting RCT in the ICU generates ethical debates*
**


During trial design and conduct, similar principles of not doing harm, protection of individuals, consent, recruitment, and ethical issues should be considered [32,33,34,35,36]. Intensive care treatments are most often invasive, and, based on real-time dynamic monitoring, the applied treatment is continuously adjusted. The critically ill population is very heterogeneous regarding patients’ profiles, combinations of organ dysfunction and responses to treatment. Thus, it is sometimes difficult to define a more homogenous group of patients to include in the controlled studies. Also, frequently, patients are excluded based on rigid exclusion criteria, and this is why, for some medications or treatments, the administration in critical care is extrapolated from non-critically ill patients.

Ideally, ICU care should be based on evidence that comes out of rigorous RCTs. But individualised practice is far from ideal, and there is a large variability. One of the ethical problems with trials conducted in the ICU is that they involve the inclusion of the individual patient in a study arm (clustered care), while clinicians individualise care. From this point of view, the results of trials might not fit well with clinical practice. Other questions regarding the critically ill recruited in a trial, as these patients have high risks of death, are: what happens during the conduct of a study when evidence builds in favour of one arm? Or should investigators not look at the data until the study is finished in order to achieve power? [10,33]. From these perspectives, large-scale randomised controlled trials are difficult to perform in ICUs worldwide. In general, to demonstrate limited effects, a high number of patients need to be included, which is a limitation.

From the methodological point of view, it is essential that the clinical trial design includes: the scientific significance, power and sample size, the outcome measures, the study arms that should be clearly defined, as well as monitoring and stopping rules [9,12,13,16,27]. Protection of human subjects implies careful monitoring for adverse events during the implementation phase. In trials including rescue therapy, only the timing of treatment, but not treatment efficacy, can be tested [33]. In ICU trials, allocation concealment is sometimes impossible, which is one of the limitations of trial methodology [37]. Other concerns are the conduct of small sample size trials, the sometimes lack of blinding possibilities and selective outcome reporting bias. Moreover, the control groups should receive standard or usual care, but this is sometimes difficult to characterise in ICU practice [4,9,13,14,16].

Due to the above, some ICU trials have only been published as research projects but have not been implemented into practice [37].

Adherence to reporting guidelines for the type of study that is conducted (for example, CONSORT for clinical trials) [79] is useful to demonstrate that the study protocol is sound and major steps are not skipped. This has large applicability in biomedical research.

XII.
**
*Ethical challenges regarding ICU research publication are similar to those involving non-critical patients*
**


Theoretical papers on ICU research ethics mention that the dissemination of results is mandatory and a duty towards society [1,14,16]. Even if results do not demonstrate positive findings, negative results should be published, together with potential treatment-associated harms, as treatment complications are not infrequent in the ICU. There are no regulations yet regarding the publication of results on media platforms before being peer reviewed and published in scientific journals. Such publications could be regarded as a double-edged sword: results might be beneficial, or the medication could be rather harmful. This tendency to apply unverified treatments (according to the scientific rigour of randomised controlled trials) has emerged as a rush during the COVID-19 pandemic and demonstrated the importance of peer review and having a sound scientific study protocol when treatments are applied to ICU patients. Peer review is a general safeguard for patients and is proof of independent confirmation of the correctness of the studies. However, the papers included in this review do not mention the moral duty of ICU researchers to perform peer review, except that peer reviewers and editors must recognise conflicts of interest [16].

XIII.
**
*Special ICU populations*
**


The conduct of studies involving ICU staff poses ethical difficulties. ICUs are demanding, can cause doubt, anxiety, and emotional distress among healthcare professionals, leading to burnout and high turnover rates [68]. Conducting research in the ICU might be a supplementary stressful factor for clinicians.Pediatric ICU (PICU) research presents distinct ethical challenges. The high-stress nature of PICUs, combined with the complexity and urgency of cases, as well as the intense emotions of parents, makes obtaining meaningful informed consent particularly difficult. Therapeutic misconception, with parents feeling pressured to consent to research under such conditions and believing it to be the best option for their child [42,43], represents one of the potential difficulties. Parents’ consent and children’s assent are difficult to obtain in emergencies. Deferred consent models might allow research to proceed easier without delaying urgent care. Some authors suggest that informed consent should be ‘appropriate’ rather than ‘fully informed’ [62]. This approach remains ethically debatable, as tensions may arise when clinicians prioritise the perceived clinical benefits of rapid testing, overriding parental autonomy in decision-making [63].Research on organ donors and organ harvesting might be hampered by inconsistencies that exist in end-of-life practices throughout the globe, as demonstrated in the ETHICUS-II trial [80]. Controlled donation after circulatory death presents significant ethical challenges in balancing end-of-life care with organ donation.Research on the process of death, dying and end-of-life approaches is hampered by variability in practices. In resource-limited settings, financial constraints drive increased ICU admissions as a revenue-generating strategy, resulting in the unnecessary use of critical care resources in cases where patient-centred approaches in palliative care could provide a more ethical alternative [70]. Second, end-of-life practices are highly variable among countries [80]. Still, the inclusion of imminently dying and recently dead populations in prospective research is necessary to improve care of dying patients [55].

XIV.
**
*Special ICU settings*
**


Research on clinical ethics is challenging as there is an overlap of research ethics and clinical ethics, starting from the fundamental ethical principles. Ethics consultations are important in the evaluation of complex ethical decisions, but from a research perspective, they come with various challenges, including methodological limitations [69].The integration of genomic research into neonatal and PICUs introduces significant ethical complexities. The emotional distress of family and staff is associated with genetic testing. Parent-child bonding might be influenced when early diagnosis of genetic conditions is revealed, leading to questions of when and how such information should be disclosed [40]. Accessibility to genomic testing and distributive justice in resource allocation topics arise in genomic testing [40,41].Emergency ICU research presents specific ethical and logistical challenges, primarily due to the urgency and complexity of critical situations [44,45]. The more severe the illness, the less time there is to conduct research, and the quicker therapeutic measures must be applied [12]. It is necessary to institute treatment immediately and not delay clinical care for research purposes. This imperative, which respects patient beneficence, leads to recruitment difficulties. Clinicians who might hesitate to use deferred consent and retrospective consent might place additional burdens on families. Extensive planning is essential to overcome logistical barriers and to ensure adherence to research protocols [45].Differences in ICU organisation, infrastructure, and practices create inequities, as not all ICU patients receive the same standard of care [72]. There is wide variability regarding adherence to best practices and the use of innovative therapies, leading to variations in performance [3,72]. Physicians may also rigidly adhere to standardised protocols, due to potential concerns of litigation for non-delivery, which might influence research protocol adherence [2].ICU costs could also be taken as research themes. Intensive care treatments place a significant financial burden on both the healthcare system and individuals, leading to accessibility and cost-effectiveness challenges [71]. ICU resource utilisation is often excessive, particularly at the end of life [70].Studies conducted during a pandemic should have the same methodological standards as those conducted in non-pandemic settings [13]. Healthcare services are overwhelmed, and real-time analysis is required to understand pathology, efficient treatments and outcomes. ICU research during pandemics faces ethical and logistical challenges, including delayed ethical approvals, triage, and resource allocation dilemmas. Despite the urgency of gathering data, ethics review processes create substantial delays, with an estimated delay of up to six months [65]. Therefore, timely approved protocols and rapid reviews are essential for pandemic preparedness. Inefficient global coordination of research efforts led to delayed data sharing and duplication of studies, as seen in the recent crisis, highlighting the need for improved strategies [67]. Since time is limited, there is insufficient guidance for both clinicians and researchers, exacerbating moral distress as resources are insufficient [64].

Are these special populations and settings similar to the general ICU or to the general intensive care patient regarding research particularities during study design, recruitment, informed consent, data collection, researcher-participant interactions, data analysis, and publication? Or are these particularities specific to these special populations and settings?

The strength of our study is its rigorous design, with key results on controversies, challenges or difficulties in ICU research being debated and thus validated, each in several published papers. However, the manuscripts have been selected by authors based on the inclusion and exclusion criteria, as well as personal experience. Thus, we admit that we might have skipped or overlooked some of the relevant papers, as some of the manuscripts, with varying classifications, have been published in critical care journals and other medical journals with a broader audience.

Maintaining the quality of the research study is a duty of each member of the research team and of those caring for the critically ill patients. Authors argue that there is a lack of guidelines for the ethical conduct of research in the critically ill and that researchers in critical care have to tackle ethical dilemmas by themselves alone as individual researchers [1,4]. All members of the ICU team participate in the research culture that allows studies to be conducted. Nurses play an important role in the recruitment process, identifying eligible patients, introducing research teams and caring for enrolled patients, but they may face barriers that might hamper study implementation [77]. It is important that ethical issues are analysed and guidelines for the ethical conduct of research in the ICU are designed to prevent unethical research. Excessively restrictive regulations might hamper obtaining ethical committees’ approval, while too loose regulations might lead to the occurrence of patient harm. Thus, the identification of key ethical challenges or barriers is of utmost importance before attempts to design guidelines for research in the ICU.

## 5. Conclusions

ICU-related research is complex and challenging from a moral point of view. Significant ethical challenges arise in ICU research during the design of the study methodology and implementation phases, making adherence to research protocols difficult and hampering the study's exactness. Being a vulnerable population with limited decision-making capacity and research-associated risks, patient and surrogate consent or alternative consent methods are most often discussed and need to be considered. This systematic review gathers items related to ICU research ethics and provides a checklist of aspects that might generate ethical dilemmas. This might constitute a starting point in the design and conduct of research studies and help researchers anticipate potential barriers, improve the management of ethical challenges and avoid unethical research.

## Figures and Tables

**Figure 1 nursrep-15-00164-f001:**
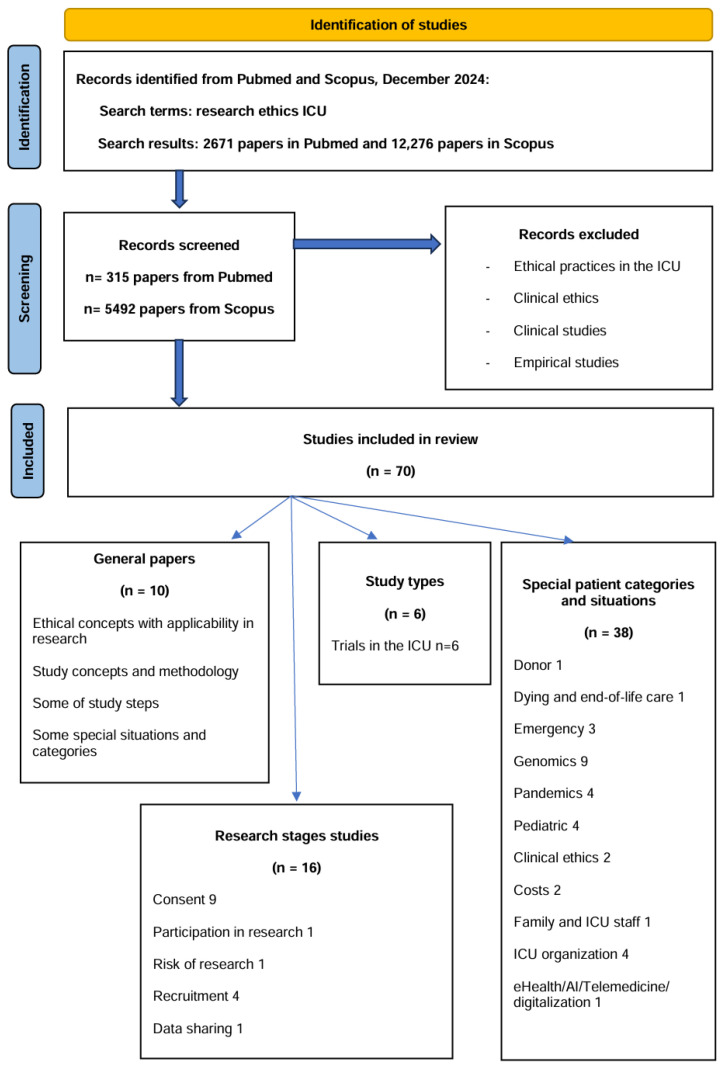
Systematic review methodology.

**Figure 2 nursrep-15-00164-f002:**
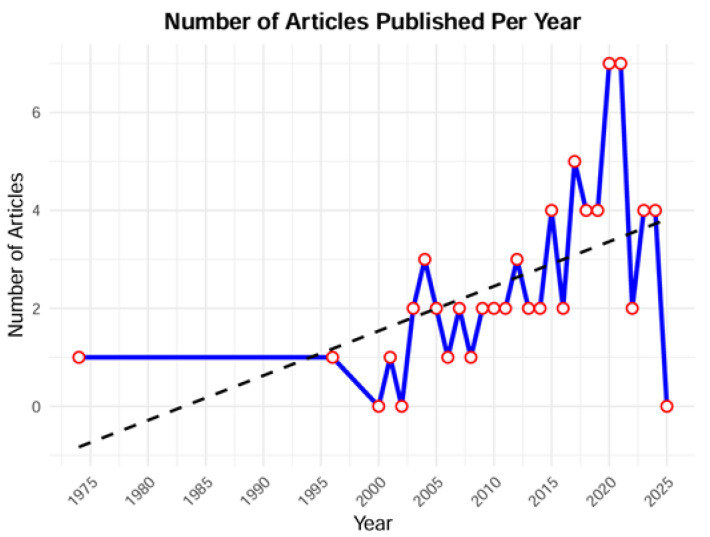
Distribution of published papers on ICU research ethics over time.

**Figure 3 nursrep-15-00164-f003:**
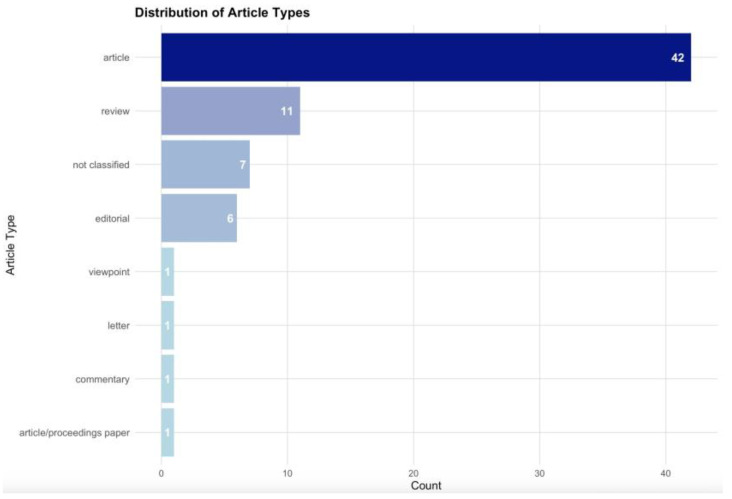
Types of manuscripts on ICU research ethics.

**Table 1 nursrep-15-00164-t001:** Ethical issues related to study steps and methodology, special populations and special settings in the ICU.

Classification of Domains	Ethical Challenge/ Controversy/Difficulty	References
I. Research ethical principles	Autonomy, beneficence, non-maleficence, justice	[1,4,8,9,14,16]
II. Study idea/ hypothesis	Relevant research for the critically ill: useful to improve clinical outcomes and to answer valuable scientific questions Test a scientific hypothesis and generate an acceptable conclusion	[1,4,9,12,14,15,17]
	Social value: the study is designed for the benefit of the community, but not of the individual research subject	[1,4,9,12,14,15,17]
	Results should be generalisable	[8]
	Clinical equipoise	[8,9,10,16,27]
	Safety and efficacy tested in a low number of patients	[4]
	Balance between benefits and harms, favourable risk-benefit ratio	[1,4,7,8,9,10,12,13,14,15,16,17,24,34]
	Protection from harm: low or minimal risk studies, reasonable risk, safeguards against anticipated risks	[4,9,10,12,13,16,20,32,35]
	Lack of guidelines for ethical conduct of research in the critically ill	[4]
	Restrictive regulations	[4]
	The quality of the study is a duty and respect for the research subject	[4]
	Scientific validity: enrollment criteria fit with the scientific goals and outcome measures reflect the purpose of the study, correct study design and protocol, sound research project	[8,15,16]
	Liability in case of harm	[1,12]
III. Researcher	Critical thinking, knowledge of ethics and law	[1,8,10,12]
	Clinical duties = best possible medical care, Research duties = carefully plan and execute the study Legal duties	[8,12,16,22]
IV. Conflicts	Conflicts of interest in the team: physician *versus* researcher, among physicians	[8,12,16,22,32]
	Logistics: the study should be performed without placing extra burdens on the clinical team	[1,38]
	Financial conflict of interest and funding: Industry involvement High costs for ICU care, high costs for research	[1,9,16]
	Researcher conflict of interest: promotion, fame, prestige	[4,14,16]
V. Research ethics board	Standard requirements for protocol approval Independent review of the protocol	[1,7,8,9,12,14,16,27,36,39]
	Expert consultants who understand the problems very well	[16]
VI. Study registration	Before commencement	[16]
VII. Patient recruitment/ enrollment	Justice: Fair selection of participants: patients should be included only if the study question cannot have an answer without enrolling critically ill patients Distributive justice in resource allocation	[7,8,9,16,40,41]
	Therapeutic misconception: patients or surrogates want to benefit from research and feel obliged to enrol	[16,22,32,42,43]
VIII. Consent (informed, voluntary, autonomous, valid)	Patients’ characteristics: cognitively impaired, captive, dependent on care	[1,4,8,9,12,16,30,44,45]
	Analysis: lack of decisional capacity, too ill to understand	[4,9,12,13,14,17,22,24,26,35]
	Protection: vulnerable population, with high mortality and morbidity, unable to tolerate supplementary risks	[1,4,8,9,12,14,16,32,34,35,46]
	Respect for autonomy	[4,7,8,9,10,14,15]
	Best interest of the patient	[9,21]
	Alternative methods of consent: surrogate or proxy consent Emotional distress, anxiety Lack of understanding Family members' disagreement	[4,7,8,9,12,13,14,17,22,23,25,26,27,28,35,36,43,47,48]
	Alternative consent methods: retrospective, deferred or delayed consent, waiver of consent, prospective (advanced planning) consent	[4,13,16,22,24,27,44,49]
	Providing information (and the quality of the information) about the study to the research subjects	[18,19,31]
	Able to withdraw from the study	[8,12]
	Documentation of consent	[12]
IX. Confidentiality	Personal data and biological samples protection	[14,15,16,29,42,50]
X. Observational studies	Registries, audits, prospective or retrospective (e.g., observational cohort, case-control studies)	[12,13,16]
XI. Interventional studies	Clinical trial design: scientific value, selection criteria, power and sample size, outcome measures, study arm allocation, allocation concealment Protection of human subjects: monitoring adverse events, stopping rules	[9,12,13,16,27,32,37]
	Control group: standard of care? (Broad variations in practice)	[4,9,13,14,16]
	Minimisation of risk, minimal risk, no more than minor increase above risk, “rescue” therapy trials	[10,33]
XII. Publication	Dissemination of study results is mandatory	[1,14,16]
	Peer reviewers and editors must recognise conflicts of interest	[16]
XIII. Special populations	Pediatric	[12,16,40,42,43,46]
	Staff and families	[15,51]
	Donor	[52]
	The dying patient	[4,53,54,55,56,57,58,59]
XIV. Special settings	Emergency (cardiac arrest, traumatic injuries, myocardial infarction, arrythmias, stroke)	[4,9,12,14,44,45,49]
	Genomic	[39,41,47,48,50,60,61,62,63]
	Pandemics	[13,64,65,66,67]
	E-health	[38]
	Clinical ethics	[68,69]
	Costs	[70,71]
	ICU organisation	[2,3,72,73]

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
