# Peer review of "Research Ethics Challenges, Controversies and Difficulties in Intensive Care Units—A Systematic Review of Theoretical Concepts"

_nursrep, 2025, doi:10.3390/nursrep15050164_

Round 1
Reviewer 1 Report
Comments and Suggestions for Authors
While there are a fair number of published papers dealing with research projects in the ICU, this paper is novel and useful in constituting a systematic review which discerns themes running through the host of particular papers. These themes are set out in detail in Table 1 and in the Discussion section.
The methodology of this study is carefully explained (see lines 75-127). There is meticulous documentation of themes pulled out from the published literature by way of frequent and extensive reference notes in Table 1 and throughout the Discussion section. In these ways the scientific soundness of the claims made is established.
The content of the paper is very compact and dense. In reading the Discussion section, I found myself wanting more expansive explanations of some points made and the inclusion of concrete examples. Permit me to present some examples of what I have in mind:
- Lines 277 – 78 “Other conflicts [of interest] might arise between different members of the team, different medical and surgical specialties, or between family and the medical team.” – Give some concrete examples here, which will also serve to elucidate the point being made.
- Lines 283-294 “A challenge for researchers is that independent REBs ask for different requirements for protocol approval.” How so? Give examples of some differences.
- Lines 317-329 “A bias during the recruitment phase could be therapeutic misconception, when individuals or relatives believe that the individual patient could have a therapeutic benefit from research. Such situations could be avoided by correct disclosure of information during the informed consent process.” Give an example of a case in which “therapeutic misconception” was operating. And give some sample language to be used during the informed consent process which avoids “therapeutic misconception.”
- Lines 355-358 “The most frequent practice in ICU research is to ask for permission from a proxy or surrogate, for example, the family in most legislative systems. However, the family members might suffer from emotional distress and anxiety when a family member is critically ill and might not be able to comprehend the information.” A concrete example would be helpful here. Lines 358-459 “Also, it seems that the proxy’s ability to predict patient preferences are highly variable.” Again, some concrete examples are in order here.
THESE ARE SAMPLES OF WHAT I HAVE IN MIND, NOT AN EXHAUSTIVE LIST.
It would take some time, and it would expand the length of the paper, to go through the Discussion section carefully elaborating on points and adding illustrative examples. But I believe the end result would attract more readers and cause the paper to be used more frequently and effectively in planning research projects in the ICU. What I suggest is an enhancement of a paper that is already very substantive and strong in content.
Author Response
Thank you very much for the review of our paper, we have responded to some suggestions:
For lines 277-78 we have added a quotation of findings, highlighted in red: For instance, in non-consensual research, the research team and the family might have different views regarding the enrollment of a patient in a research study [Moore_2004]. Also, the research study might be perceived as an extra burden on the already difficult task of caring for a critical patient and the recruitment process might be hampered [Krewalk_2024].
Line 283-294- we provide the example of genomic research challenges, where differences lead to delays in the approvals and implementations of studies: The approval processes of local and national ethics committees need to be harmonized through a common protocol, especially for multi-national genetic studies [39- Tridente].
Lines 317-329- we have added a concrete example here: For instance, the parents of a very sick child with limited chances of survival might be inclined to confirm participation in research, hoping that the new therapy might cure the disease. The research team needs to inform the family in an honest way that no evidence is in favor of the new treatment and this is why the study is in fact conducted. Correct disclosure of information during the informed consent process is very important so that no false hopes are induced.
Lines 355-358- for these we have cited a paper on the analysis of the concordance between the surrogate and patietns wishes to participate in research, from Chest 2001: There might be false positives and false negatives when the proxy appreciates the critical patients’ wish to participate in clinical research and it seems that these assumptions are not better than tossing a coin [Coppolino].
We have also brought some clarifications:
Line 209: In fact, the moral principle of beneficence, together with the opposite non-maleficence, might themselves generate a conflict. These concepts apply to both the society or larger group of critically ill patients, but also to the individual patient that becomes the study subject and might suffer from harm as a result of study risk. In research, a potential social value is to be compared to individual risks.
Line 548: All members of the ICU team participate in the research culture that allows research to be conducted. Nurses play an important role in the recruitment process, identifying eligible patients, introducing research teams and caring for enrolled patients, but they may face barriers that might hamper study implementation [Krealk].
Thus, we have added in total three extra references:
- HALL, K.; MOORE, A.; HICKLING, K. Critical care research ethics: making a case for non-consensual research in ICU. Critical Care and Resuscitation, 2004, 6.3: 218-25.
- KREWULAK, Karla, et al. ICU Care Team’s Perception of Clinical Research in the ICU: A Cross-Sectional Study. Critical Care Explorations, 2024, 6.4: e1072.
- COPPOLINO, Michael; ACKERSON, Lynn. Do surrogate decision makers provide accurate consent for intensive care research?. Chest, 2001, 119.2: 603-612.
Reviewer 2 Report
Comments and Suggestions for Authors
Dear authors,
this is a very good article that presents a very well comprehensive overview of the topic, which is a very interesting one.
There is some room for improvements:
1) methodology: please elaborate more clearly the inclusion and exclusion criteria. A table might be useful here.
Also, please elaborate more clearly on how the quality of the included articles was assessed, if it was. If no formal quality assessment was done, this should be made explicit. A Table for the assessment might be useful too.
2) table 1: it is very comprehensive but you might consider simplifying it by summarizing a bit.
3) Ackonelwding limitations to the study would be good.
4) minor typographical edits: you repeat complex twice at line 49-50 and other minor typos.
5) Though the article is well written from a formal point of view, I found the Introduction to be not so effective. Especially I found the paragraph starting at line 62 not very well written / structured. Please consider reviewing it.
Author Response
Thank you very much for the appreciation of our manuscript.
Inclusion and exclusion criteria paragraph was rephrased and we hope that it is more clear now: We excluded empirical studies, research based on observation and experience, and papers on clinical ethics, if not discussing aspects related to research ethics in the ICU. We included in this PRISMA-guided systematic review theoretical papers with argumentative patterns and other papers discussing challenges, difficulties or controversies related to the ethical conduct of research studies in the ICU. No formal quality assessment was performed, as the included papers are theoretical ones.
We consider that extra tables would not bring more information compared to the text. Also, we consider that Table 1 is comprehensive and we would prefere to maintain it like this. Surely, it is long, but the reader might use it like a checklist. Please do not hesitate to let us know if you consider that we should really change it.
We have added limitations paragraph: The strength of our study is a rigorous design, with key results on controversies, challenges or difficulties in ICU research being debated, thus validated, each in several published papers. However, the manuscripts have been selected by authors based on the inclusion and exclusion criteria, as well as personal experience. Thus, we admit that we might have skipped or overlooked some of the relevant papers, as some of the manuscripts, with varying classifications, have been published in critical care journals and other medical journals with a broader audience.
We have addressed the typographical edits you pointed out, as well as others, thus, we have made several minor corrections and they are highlighted in red.
We have reviewed the paragraph at line 62 and it is marked in red: There is a general tendency to introduce procedures to clinical practice without rigorous evaluation in critical care [9]. Is one of the reasons for this the difficulty to conduct rigorous studies? Truog et al. stated that „if current trends continue, within several years it could become nearly impossible to conduct research in critical care medicine”.
Reviewer 3 Report
Comments and Suggestions for Authors
This is an excellent paper filling a gap in the bioethical literature covering moral issues that arise in Intensive Care Units. I think it should be published as is; some transitions could be smoother in Section I, but they do not sufficiently interfere with clarity to warrant a change. The paper made some surprising discoveries; I was not aware to so little work has been done on the ethics of the ICU. The section covering newer moral issues that arise, such as genomics, is strongly needed, and I am pleased that the authors included it. I trust that your article will have a wide readership and encourage further research in the ethics of the ICU.
Author Response
Thank you very much for the work with our manuscript and your appreciation.